# Antiretroviral therapies and status of people living with HIV in Japan: An update from hospital survey and national database

Yoshiyuki Yokomaku[1][☯], Tatsuya Noda[2][☯]*, Mayumi Imahashi[1], Yuichi Nishioka[2], Tomoya Myojin[2], Aikichi Iwamoto[3], Tomoaki Imamura[1]

1 Department of Infectious Diseases, NHO Nagoya Medical Center, Nagoya, Aichi, Japan, 2 Department of Public Health, Health Management and Policy, Nara Medical University, Kashihara, Nara, Japan, 3 Office of Project Management, Japan Agency for Medical Research and Development, Chiyoda-ku, Tokyo, Japan

☯ These authors contributed equally to this work.
* noda@naramed-u.ac.jp

## Abstract

No updated data on people living with HIV (PLHIV) in Japan have been available since 2015, leaving a critical gap in understanding the current status of care and treatment. Therefore, this study aimed to conduct a nationwide evaluation of the second and third goals of the "90-90-90 target" defined by UNAIDS between 2016 and 2020. The study utilized data from approximately 360 core hospitals through structured questionnaires and the National Database of Health Insurance Claims and Specific Health Checkups (NDB). Key findings revealed that over 95% of diagnosed outpatients were retained in care (second 90), and more than 99% achieved successful viral suppression (third 90). A significant transition to single-tablet regimens and newer, highly effective antiretroviral drugs was observed, optimizing treatment adherence and outcomes. These results underscore the efficacy of Japan's universal health insurance system in ensuring consistent access to HIV care and treatment, supporting both individual patient outcomes and national surveillance efforts.

## Introduction

Antiretroviral therapy (ART) contributes to improved prognosis and reduced transmission of HIV among people living with HIV (PLHIV) [1]. In 2016, the Joint United Nations Program on HIV/AIDS (UNAIDS) set the 90-90-90 target for 2020: 90% of the infected persons knowing their HIV status, 90% of them receiving HIV treatment, and 90% of people receiving treatment with successful viral suppression. More recently, the 95-95-95 target was established for 2025, and improving the quality of life (QOL) of PLHIV has been added as an important goal [2].

No statistics on PLHIV epidemiological status in Japan have been released since the report by Iwamoto et al. in 2017, who presented information updated to December 2015 [3]. Although the third UNAIDS goal had already been achieved by that time, the first and second goals had not been reached: 85·6% of HIV-positive cases were diagnosed, and 82·8% of those

**Data Availability Statement:** There are legal restrictions on sharing de-identified data sets. Japanese law relating to NDB prohibits the public

release of NDB data unless special permission is granted by the Ministry of Health, Labor and Welfare. The data for this study and the public release coordination to the Ministry of Health, Labor and Welfare are being conducted by the following data management organization: 'NDB Research Data Management Committee' email: g_phdata@naramed-u.ac.jp.

**Funding:** This work was supported by Health Labour Sciences Research Grants 20HB2001 (acquired by YY), 20HB1001 (acquired by TN), and 23HB1001 (acquired by TN) [funder URL: https://mhlw-grants.niph.go.jp/] and by JSPS KAKENHI Grant JP20H00623 (acquired by TN) [funder URL: https://www.jsps.go.jp/english/e-grants/]. The funders played no role in the study design, data collection, data analysis, interpretation, or manuscript writing.

**Competing interests:** The authors have declared that no competing interests exist.

diagnosed were under sustained anti-HIV therapy. Even when these numerical data are encouraging, the proportion of core hospitals, which were designated by Ministry of Health, Labour and Welfare and provides ART according to the guidelines [3], responding to all queries was 80·6% (308/382), and the number of PLHIV cases among foreign citizens living in Japan was based on an estimate. Moreover, the aforementioned study did not address the composition of anti-HIV therapies administered to PLHIV who are retained in care in the core hospitals.

Japan achieved universal health coverage in 1961. Nearly all Japanese citizens are covered by some kind of medical insurance [4]. Since 2008, all claims from medical services providers have been integrated into a large national database called the National Database of Health Insurance Claims and Specific Health Checkups of Japan (NDB) [5]. All medical institutions use the same claims forms to obtain reimbursements from insurers.

Almost all Japanese patients diagnosed with HIV infection are currently undergoing anti-HIV therapy, mainly provided at core hospitals for AIDS treatment, where specific physicians, previously screened and designated by the local government authorities, are in charge of PLHIV medical care under the social security system. These core hospitals are distributed across Japanese prefectures and vary largely in the number of HIV/AIDS outpatients assisted. The use of large-scale claims data tends to overcome this limitation and enables a more accurate assessment of the current AIDS situation in Japan.

Using a combination of HIV-related disease name assignments and anti-HIV drug prescription records, highly accurate data on HIV-infected persons and comprehensive epidemiological and medical information can be obtained from the NDB. Furthermore, the NDB would allow the grouping of HIV-infected people in Japan into temporal cohorts to compare their medical evolution.

The main objective of this study was to update information regarding the second [people with diagnosed HIV will receive antiretroviral therapy (ART)] and third (people receiving ART will have viral suppression) proportions of the 90-90-90 target in Japan by analyzing data from 2016 to 2020. To this end, we used information from two sources: core hospitals distributed across 47 Japanese prefectures and data extracted from the NDB. Through this innovative approach, we aimed to identify the achievements and challenges of the Japanese medical system over the next few years to improve the QOL of PLHIV and reach UNAIDS goals.

## Methods

### Core hospitals for HIV/AIDS treatment in Japan

On July 28, 1993, the Ministry of Health and Welfare (now the Ministry of Health, Labour and Welfare) issued the "Establishment of Core Hospitals for AIDS Treatment" directive to each prefecture. This marked the beginning of efforts to establish designated hospitals where patients with HIV infection and AIDS [hereinafter referred to as "people living with HIV" (PLHIV)] could receive medical care free from discrimination and prejudice. Medical institutions with advanced medical capabilities within each region were selected for this role.

Following the settlement of the drug-related AIDS lawsuit in March 1996, the AIDS Clinical Center (ACC) was established on April 1, 1997, at the National Center for Global Health and Medicine Hospital. Subsequently, on April 25, 1997, the directive "Development of Regional Block Core Hospitals for AIDS Treatment" was issued, leading to the establishment of regional block core hospitals (block centers) in eight regions across Japan. Later, on March 31, 2006, in line with the 2006 revision of the AIDS Prevention Guidelines, the directive "Development of Core Center Hospitals for AIDS Treatment" was issued, and core center hospitals were designated from among the core hospitals in each prefecture. Since their inception, core hospitals

have served as the cornerstone of AIDS treatment in Japan. As of the end of 2021, a total of 377 core hospitals had been designated across the country.

## Information collected from core hospitals

We designed a questionnaire to be answered by administrative and healthcare staff (physicians, nurses, etc.) of core hospitals distributed across Japan. The questionnaire included the following items:

1. Number of outpatients with known HIV/AIDS diagnosis at the first hospital visit.

2. Number of outpatients with HIV/AIDS untreated at the first hospital visit.

3. Number of outpatients diagnosed after their first visit to the hospital.

4. Number of outpatients with HIV/AIDS retained in care after diagnosis (including patients undergoing ART and patients under follow-up, without receiving ART).

5. Number of outpatients with HIV/AIDS on treatment.

6. Number of HIV/AIDS outpatients virally suppressed (successfully treated).

7. Number of outpatients with HIV/AIDS whose nationality was not Japanese.

8. Number of outpatients with HIV/AIDS who died due to any cause.

9. Number of outpatients whose CD4 count at HIV/AIDS diagnosis was less than 200/μL.

10. Number of outpatients clinically considered as cases of treatment failure, with $\geq 200$ copies/mL in two consecutive viral load measurements.

The survey was conducted in 47 Japanese prefectures. The administrative staff in each prefecture distributed the questionnaires, collected the completed questionnaires, and sent them back to the study group. Only core hospitals that responded to at least Items 4), 5), and 6) were included in this analysis. The following definitions were considered:

- Newly untreated PLHIV: HIV-infected persons that have never been treated at the time of their first visit to a core hospital.

- Diagnosed patient: HIV-positive person by both screening and confirmatory HIV testing.

- Patient retained in care: HIV-infected person linked to core hospitals after diagnosis, whether on treatment or not.

- Patient on treatment: HIV-infected person under sustained treatment

- Patient virally suppressed: An HIV-infected person under sustained treatment and a viral load clinically judged as suppressed or objectively shown to not exceed 200 copies/mL in at least two consecutive determinations.

- Treatment introduction rate: Number of outpatients who started ART (for the first time) or resumed ART/number of outpatients retained in care.

- Treatment success rate: (number of patients under sustained ART − number of patients who objectively failed to respond to ART)/number of patients under sustained ART.

After performing a global-scale analysis, we analyzed the parameters 4), 5), and 6) separately for nine Japanese geographical regions (Hokkaido, Tohoku, Kanto-Koshinetsu, Metropolitan, Hokuriku, Tokai, Kinki, Chugoku-Shikoku, and Kyushu). Thus, the eventual

influence of demographic variables such as population density and access to medical facilities could be detected.

Hospitals not reporting the number of patients retained in care after diagnosis, the number of patients on treatment, and the number of patients that were virally suppressed were excluded from our analysis.

In Japan, the AIDS Trends Commission collects data submitted annually by physicians who have diagnosed patients with HIV/AIDS for the first time. Thus, this commission is in charge of the active surveillance of HIV/AIDS cases in Japan and, based on the collected data, communicates the actual trends of some epidemiological parameters through the AIDS Trends Commission Report (ATCR). Physicians are required to report these data in the context of the Infectious Disease Control Law [6].

### Information collected from the NDB

The NDB is one of the largest medical-visit databases in the world and contains information on the annual medical visits performed by approximately 110 million Japanese citizens. Its coverage of nearly the entire Japanese population ($\sim$ 127 million persons), along with wide accessibility to medical care in Japan, results in minimal biases and omissions. We obtained permission from the Japanese government to use the NDB data in May 2022, and after receiving the data on October 30, 2022, we identified PLHIV receiving treatment based on claims data. In this dataset, the anti-HIV drugs prescribed for PLHIV were recorded. Patients who were prescribed anti-HIV drugs (see S1 Table) at least once were defined as PLHIV receiving treatment, and we identified all the prescription patterns these PLHIV received during the observation period. Based on the S1 drug list, we classified the antiretroviral treatments according to the prescribed active ingredients, classes, and formulations. The combination of the top ten ART prescriptions in each prefecture allowed us to establish the top ten regimens for Japan. To extract the number of individuals "retained in care" and "on treatment" from the NDB, we used the codes 160163650 and 160225550, previously established by the Ministry of Health, Labour, and Welfare of Japan. Both PLHIV categories were defined as having visited a hospital or clinic at least once within a 6-month period.

The National Hospital Organization Review Board for Clinical Trials in Nagoya and Nara Medical University Ethics Committees reviewed and approved this study (approval numbers 2016–86 and 2831, respectively) and waived the requirement of obtaining informed consent as the article does not include any participant-identifying data.

### Statistical analysis

The numbers of PLHIV retained in care and on ART from 2016 to 2020 were obtained from the NDB. These values were compared with data extracted from the core hospitals' questionnaires. STATA version 14 (StataCorp LP, College Station, TX., USA) was used to analyze the data. To map the core hospitals' locations and calculate the coverage rates of the top ten regimens identified, we used ArcGIS Desktop ver. 10.7 (ESRI Corp., Redlands, CA, USA).

## Results

### Coverage of core hospital surveys

From 2016 to 2020, we obtained responses from all the core hospitals. Of them, at least 94–96% of the existing core hospitals (~380) are included in this analysis using the method described in the Method part (Table 1). These hospitals were responsible for approximately 22,000 (2016) to $\geq$ 27,000 (2020) PLHIV outpatients, and at least 97% of them retained in care

**Table 1. Coverage of the survey conducted.**

| Year | 2016 | 2017 | 2018 | 2019 | 2020 |
|---|---|---|---|---|---|
| Total number of core hospitals | 382 | 381 | 382 | 380 | 379 |
| Number of core hospitals included [a] | 362 | 362 | 362 | 361 | 365 |
| **Ratio** | **0·948** | **0·950** | **0·948** | **0·950** | **0·963** |
| Total number of patients retained in care | 22,516 | 24,022 | 24,022 | 25,988 | 27,189 |
| Number of patients retained in care included [b] | 22,050 | 23,323 | 23,323 | 25,565 | 26,452 |
| **Ratio** | **0·979** | **0·971** | **0·971** | **0·984** | **0·973** |

[a] Hospitals not informing the number of patients retained after diagnosis, the number of patients on treatment, and the number of patients that were virally suppressed were excluded from our analysis.

[b] Most patients visited the hospital for their control quarterly.

in these hospitals (Table 1). In 2020, the information collected from 365 facilities (96·3% of the target facilities) showed that 26,452 outpatients (97·3%) were retained in care in core hospitals. Only a small proportion of outpatients, corresponding to 14 facilities, were excluded from the analysis in 2020.

We limited our analysis to those core hospitals that provided information for the following three items, at least: (a) the number of outpatients retained in care after diagnosis, (b) the number of outpatients on treatment, and (c) the number of outpatients successfully treated, as defined in the "Methods" section.

Fig 1A shows the location of the core hospitals assessed in 2020; these hospitals were responsible for the treatment and follow-up of PLHIV across all Japanese prefectures. Fig 1B indicates the global volume of HIV patients in each prefecture grouped into five quantitative categories. The placement and number of hospitals were in accordance with the medical plans established by each Japanese prefecture. We detected great heterogeneity in the number of facilities and patients among prefectures, with thousands of PLHIV attending hospitals in the largest cities (e.g., Tokyo, Osaka, Nagoya, Fukuoka), whereas fewer than 100 patients attended these facilities in regions such as Tohoku and Shikoku. Fig 1C illustrates the trend toward the consolidation of medical care for PLHIV in facilities receiving a high number of patients. Thus, in 2020, only 13·4% of these core hospitals were in charge of $\geq$ 100 retained patients, comprising 76·9% of the total number of retained patients (Fig 1C).

The numbers of annual deaths and non-Japanese PLHIV visiting core hospitals in 2016–2020 are shown in Fig 2A and 2B, respectively. Although the number of PLHIV enrolled in core hospitals during 2016–2020 increased steadily (Table 1), the number of all-cause deaths remained nearly constant, with approximately 130 annual deaths up to 2019, after which it increased to 175 annual deaths (Fig 2A). This indicates a death rate (all-cause death) of 0·6% in 2020. In contrast, the number of non-Japanese PLHIV receiving medical care at core hospitals increased annually at a rate of approximately 200 and 100 per year before and after 2018, respectively (Fig 2B).

We compared the changes in the number of newly untreated PLHIV and AIDS cases according to core hospital surveys with those based on data from the ATCR for 2016–2020. The number of patients with CD4 counts below 200/μL was considered an indicator of active infection (Fig 2C). Although most parameters showed a declining trend (with the exception of an initial increase in newly untreated PLHIV), the number of newly untreated PLHIV consistently exceeded the annual count of new AIDS cases, irrespective of the data source. Moreover, the data from the ATCR were always lower than those obtained through the surveys conducted at core hospitals, suggesting underreporting. Interestingly, we found that the number of

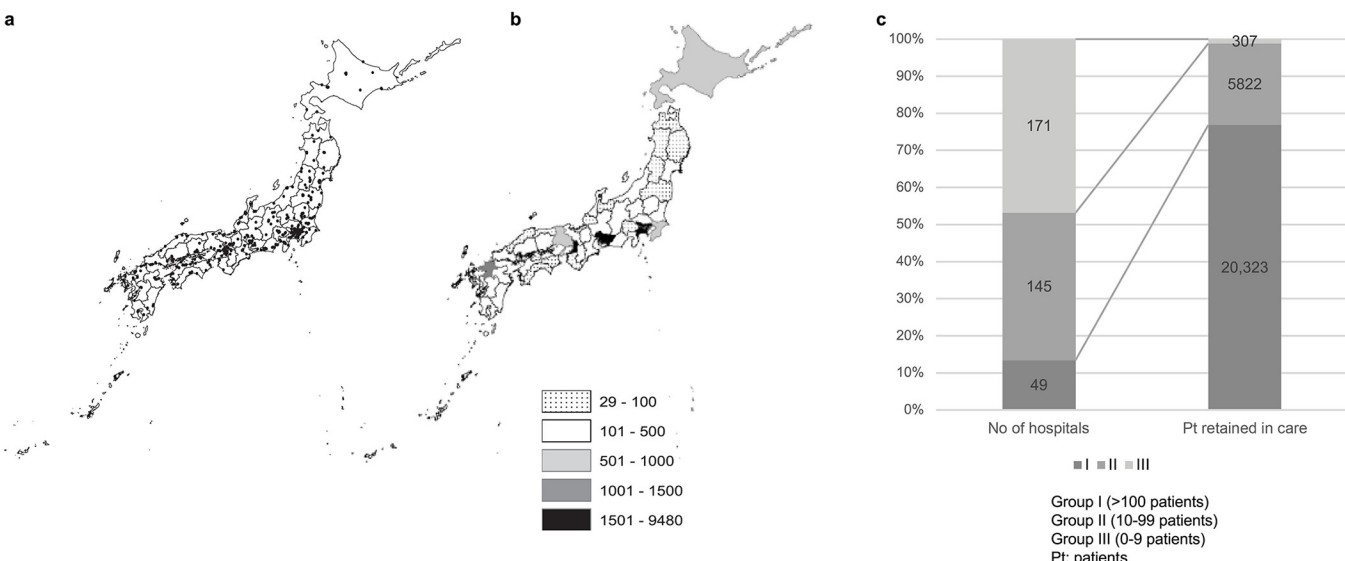

**Fig 1. Distribution of core hospitals and volume of PLHIV attending core hospitals across Japan, 2020.** The URL of the core hospitals' list is: *https://hiv-hospital.jp/map/index_kubun@shiteijiritsu.html* (accessed on Sep.12th, 2024) (a) Almost all PLHIV were assisted in core hospitals distributed across Japanese prefectures. The placement and number of these hospitals are determined by each prefecture. (b) A high proportion of the outpatients who visited core hospitals were concentrated in large cities such as Tokyo, Osaka, Nagoya, and Fukuoka, while much fewer attended core facilities in prefectures like Tohoku or Shikoku. (c) A clear trend toward the consolidation of medical care in a small number of facilities receiving a very high number of patients was detected.

patients with CD4 counts below 200/μL, which should be considered confirmatory for AIDS, consistently exceeded the rate of AIDS onset diagnoses each year. However, the number of AIDS cases recorded from the survey was closer to the number of people with CD4 counts below 200/μL.

Based on the values shown in Fig 2C, we calculated the proportion of AIDS cases/newly untreated PLHIV according to each dataset (Fig 2D). Despite the steady decreasing trend in the absolute values for the entire period (Fig 2C), the proportion of AIDS cases/newly untreated PLHIV increased from 2018/2019 onwards (Fig 2D), with a considerable gap between the ratios of reported AIDS cases/newly untreated PLHIV and persons with CD4 counts less than 200/μL/newly untreated PLHIV, which was consistently higher.

To present a global outlook on how key parameters related to PLHIV have changed over time in Japan, we compared the number of patients diagnosed, retained in care, on ART, and virally suppressed each year during 2016–2020 from the information collected through core hospital surveys (Fig 3A). The number of patients diagnosed with HIV infection increased during this period. The number of patients retained, the treatment continuation rate, and the treatment success rate also increased steadily during this period. The last two parameters are directly linked to the second and third UNAIDS goals. Thus, although the total number of out-patients covered by core hospitals surpassed 25,000 in 2019, approximately 94% of those diagnosed with HIV started ART, and ≥99% of those under ART achieved virus suppression, thereby demonstrating the success of the treatments applied.

## Regional and inter-institutional differences

We examined the number of retained/on treatment/virally suppressed patients first globally and then for the nine Japanese regions in 2020 (Fig 3B and 3C). In line with the country-level data (Fig 3B), and regardless of the population size included in each prefecture (which varied from <300 patients in Hokuriku to > 12,000 patients in the Metropolitan Area), all regions

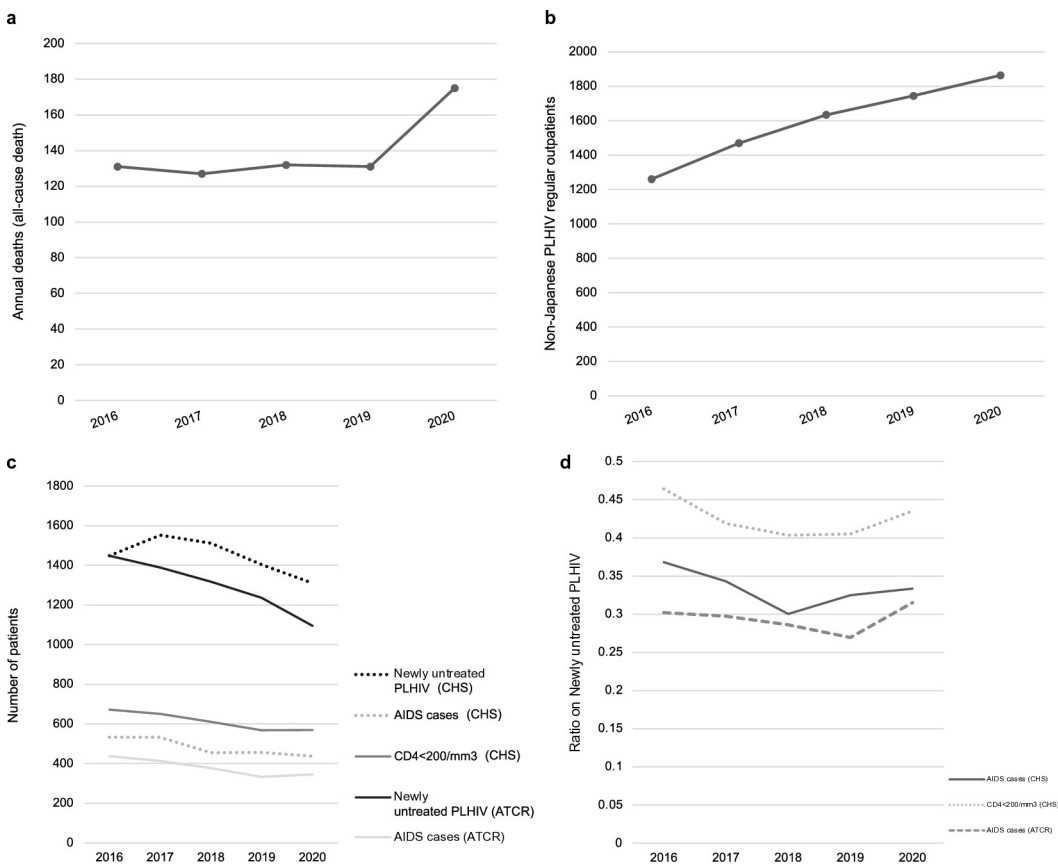

**Fig 2.** Trends in annual all-cause deaths (a) and non-Japanese PLHIV (b) receiving care at core hospitals, Japan, 2016–2020. Data source: core hospital surveys (CHS). Trends in relevant HIV infection indices, Japan, 2016–2020. (c) Absolute values; (d) ratios with respect to newly untreated PLHIV. Data source: core hospital surveys (CHS) and the AIDS Trends Commission Report (ATCR).

had a treatment introduction rate of ≥94% and a treatment success rate of ≥99%, without significant differences among them (Fig 3C).

Additionally, using data from the core hospital survey of 2020, the possible incidence of the number of retained patients per facility (≥100, 10 to 99, or 1 to 9) on the treatment-introduction and treatment-success rates was analyzed (Fig 3F–3H, respectively). The proportion of persons receiving ART with respect to those retained was over 95% across all hospital categories, and that of persons virally suppressed with respect to persons under ART was ≥99%, which corroborated that the treatment introduction rate and treatment success rate were consistently high and not influenced by patient volume.

As more patients are being treated outside the core hospital system, the NDB may be regarded as an adequate health-related data source to address important public health topics, such as AIDS. To assess the usefulness of the NDB in providing robust data on PLHIV and AIDS management, we calculated the ratio between the number of patients retained in care according to the core hospital survey and the NDB based on datasets per prefecture from October to December 2020 (Fig 3D). The same calculation was performed for the number of patients who were on ART (Fig 3E). It was observed that these datasets tended to match, with most ratios close to 1—corroborating the capacity of the NDB to provide reliable information on HIV/AIDS on a global scale.

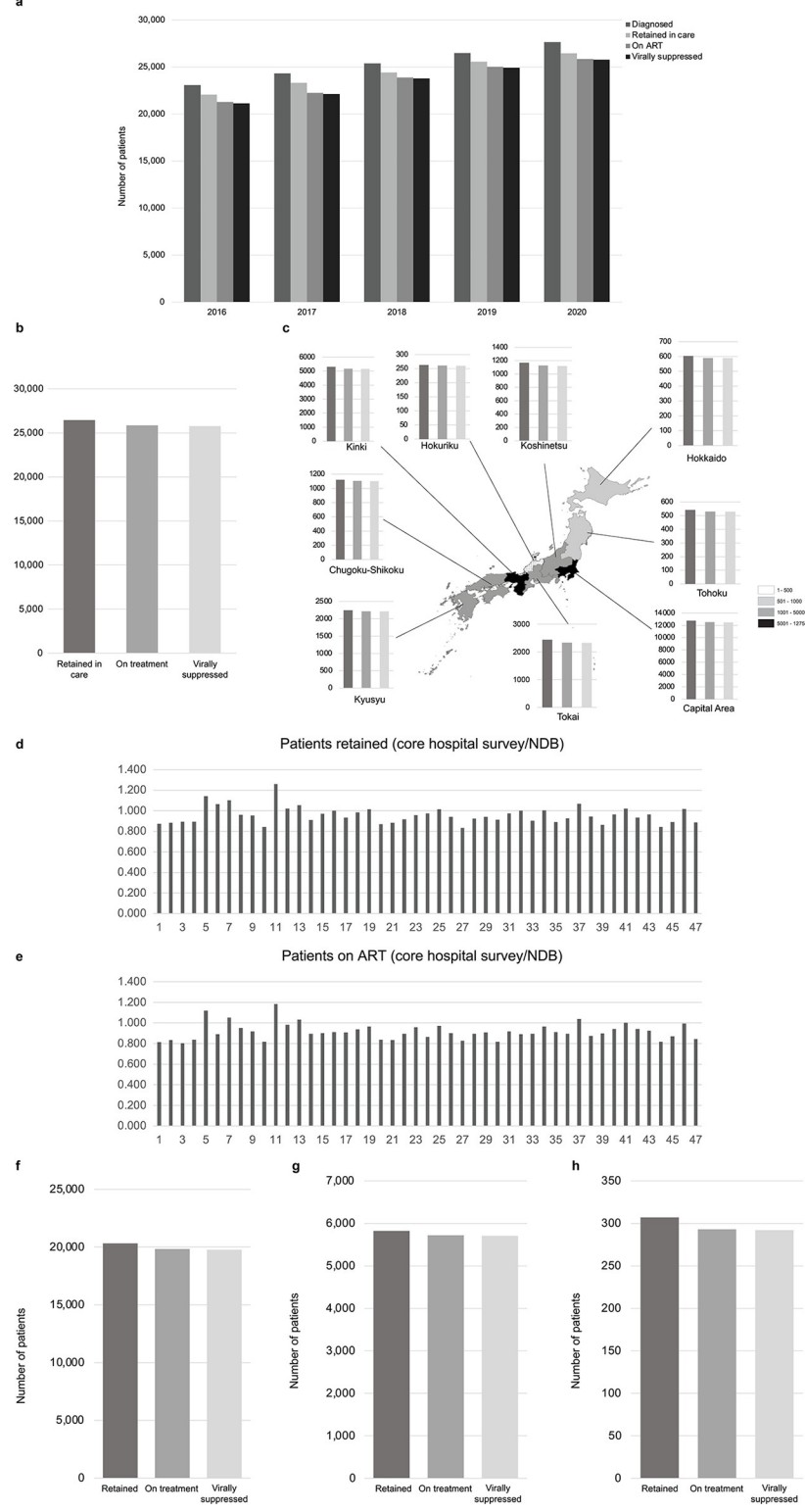

**Fig 3.** (a) Time course of parameters related to PLHIV status in Japan, 2016–2020. Data source: core hospital surveys. Parameters related to PLHIV status across the nine Japanese regions, 2020. (b) Global data; (c) regional data. The figure legend indicates the PLHIV population range in each region. Data source: core hospital surveys. Matching between core hospital surveys and NDB datasets at the individual prefecture level, October 2020–December 2020. (d) Ratio of the number of retained patients according to the NDB and the number of retained patients according to the

core hospital survey. (e) Ratio of the number of patients receiving ART according to the NDB to the number of patients receiving ART according to the core hospital survey. The numbers on the X-axis indicate the prefectures as follows: 1. Hokkaido, 2. Aomori, 3. Iwate, 4. Miyagi, 5. Akita, 6. Yamagata, 7. Fukushima, 8. Ibaraki, 9. Tochigi, 10. Gunma, 11. Saitama, 12. Chiba, 13. Tokyo, 14. Kanagawa, 15. Niigata, 16. Toyama, 17. Ishikawa, 18. Fukui, 19. Yamanashi, 20. Nagano, 21. Gifu, 22. Shizuoka, 23. Aichi, 24. Mie, 25. Shiga, 26. Kyoto, 27. Osaka, 28. Hyogo, 29. Nara, 30. Wakayama, 31. Tottori, 32. Shimane, 33. Okayama, 34. Hiroshima, 35. Yamaguchi, 36. Tokushima, 37. Kagawa, 38. Ehime, 39. Kochi, 40. Fukuoka, 41. Saga, 42. Nagasaki, 43. Kumamoto, 44. Oita, 45. Miyazaki, 46. Kagoshima, 47. Okinawa. Parameters related to PLHIV status according to the mean volume of outpatients per facility in Japan, 2020. (f) Facilities with ≥100 retained patients (n = 49); (g) Facilities with 10–99 retained patients (n = 145); (h) Facilities with 1–9 retained patients (n = 171). Data source: core hospital survey. The proportions of Fig 3(A)–3(C), 3(F)–3(H) are shown in S2 Table.

## Antiretroviral therapy: Quantitative and qualitative progress

Using data extracted from the NDB, we analyzed the content of anti-HIV therapies administered to PLHIV in Japan from 2013 up to 2020 in detail (see S1 Fig). We focused on drug families, key drug classes, and administration regimens. Since 2017, tenofovir alafenamide (TAF) has been increasingly selected as a nucleoside reverse transcriptase inhibitor (NRTI) along with lamivudine (3TC) and emtricitabine (FTC) to the detriment of other NRTIs (see S1 Fig). Thus, in 2020, 65·3% of patients were treated with TAF. Integrase strand transfer inhibitors (INSTIs) have progressively been adopted as key drugs (see S1 Fig) during this period, and the use of STRs increased steadily (see S1 Fig). By the end of 2020, the proportion of patients utilizing INSTI was 82%, and 54.7% received STRs (see S1 Fig).

Table 2 shows the evolution of the ten most frequently prescribed regimens throughout 2016–2020, according to data from the NDB. When DVY, an anti-HIV drug containing TAF, was launched, TVD was quickly replaced with DVY. Furthermore, STRs using an INSTI as a key drug tended to occupy the top positions year after year and were commonly used in 2019 and 2020. SMT, BVY, and DVT were frequently prescribed.

The adoption rate of medication schemes involving the top ten regimens across Japanese prefectures throughout 2016–2020 are illustrated (see S2 Fig). The cumulative data per year and region depicted as histograms indicate that, in 2020, ≥80% of PLHIV attending medical institutions in different Japanese prefectures received anti-HIV therapies included among the more common choices at the national scale, with STRs occupying a central role. The adoption rate of the top ten regimens increased at a higher rate in the prefectures in northern and central Japan.

**Table 2. Time-course of the antiretroviral therapies (ART) most commonly prescribed in Japan ("Top ten"), 2016–2020.**

|  | 2016 | 2017 | 2018 | 2019 | 2020 |
|---|---|---|---|---|---|
| 1 | DTG + TVD | DTG + DVY-HT | DTG + DVY-HT | DTG + DVY-HT | **BVY** |
| 2 | **TRI** | **TRI** | **TRI** | **TRI** | DTG + DVY-HT |
| 3 | DRVN_800 + RTV + TVD | **GEN** | **GEN** | **BVY** | **TRI** |
| 4 | RAL_400 + TVD | DVY-HT + RAL_400 | DVY-HT + RAL_600 | **GEN** | **GEN** |
| 5 | **GEN** | EZC+RAL_400 | DVY-LT + PCX | DVY-HT + RAL_600 | DVY-HT + RAL_600 |
| 6 | EFV + TVD | DVY-LT + PCX | DVY-HT + RAL_400 | **SMT** | **SMT** |
| 7 | EZC+RAL_400 | DVY-HT+EFV | DVY-HT+EFV | DVY-HT+EFV | **DVTs** |
| 8 | DTG + EZC | DRVN_800 + DVY-LT + RTV | DRVN_800 + DVY-LT + RTV | DVY-HT + RAL_400 | DVY-HT + RAL_400 |
| 9 | DRVN_800 + EZC + RTV | DTG + EZC | DTG + EZC | EZC+RAL_600 | DVY-HT+EFV |
| 10 | **STBs** | DTG + TVD | EZC+RAL_600 | DTG + EZC | **ODFs** |

Data source: NDB. Single-tablet regimens (STRs) are shown in bold. All regimen abbreviations are listed in S1 Table.

## Discussion

Previous studies addressing temporal changes in the medical care and treatment of PLHIV in Japan generated limited information because of different factors, including regional and sample size biases [7–9]. Unbiased sample populations, such as those covered by large-scale claims data, may overcome these limitations. The NDB covers the medical services claims of ≥100 million persons and thus constitutes the largest claims database not only in Japan but also in the world [10–12]. Previously, we developed a novel methodology to trace patient data using the NDB [13].

In this study, we identified HIV/AIDS patients on treatment by combining multiple datasets and observed a clear shift towards INSTI prescriptions, as well as the sustained replacement of tenofovir disoproxil fumarate by TAF among NRTIs. Moreover, we detected a clear trend toward single-tablet regimens at the expense of non-single-tablet regimens, which might result in better adherence [14].

This combination of efficient drugs and good patient adherence may explain the excellent treatment success rates. The NDB, which covers the vast population receiving medical care through the public insurance system, is a valuable tool for analyzing and evaluating health and welfare policies in Japan. Tanaka et al. [9] analyzed ART changes between 2016 and 2019 using data from the NDB. This open dataset provides the amount of prescribed ART but not the data on ART combinations as it does not include prescription information for each individual. Instead, the dataset we used in this research considered individual patients enabling us to track the ART combination trends.

In this study, data on PLHIV extracted from the NDB were validated based on the number of HIV outpatients reported by core hospitals, underscoring the possibility of tracking PLHIV status in Japan using data from both information sources. It is believed that the difference arises because, in some regions, the percentage of PLHIV visiting core hospitals varies across areas. This contrasts with the results of the NDB, which can track the entire PLHIV population.

Regarding the HIV/AIDS endemic, this study shows that the number of newly untreated PLHIV in Japan decreased during the study period. However, according to the World Health Organization, in East Europe, Central Asia, the Middle East, North Africa, and Latin America, the number of newly untreated PLHIV cases increased over the past ten years [15]. The WHO reported that there were 39 million PLHIV cases worldwide in 2022, and 630,000 died from AIDS-related illnesses [16]. Even when we considered all-cause death among PLHIV, it is worth noting that we recorded a death rate of 0·6%, which is nearly half of that occurring at a global scale. This may be due to better access to healthcare in Japan.

Furthermore, this study reveals that Japan has already achieved the second and third goals of the 95-95-95 UNAIDS ideal. Notwithstanding, the first goal–95% of infected persons knowing their HIV status–could not be addressed because we were unable to obtain the estimated number of HIV-infected people. Achieving this goal remains a significant challenge in Japan due to limitations in the HIV testing system, which directly impact the evaluation and outcomes. Various methods are currently being considered to estimate the number of infected individuals.

Our research demonstrates that by combining data extracted from the NDB with data from other sources, it is possible to perform a comprehensive and highly reliable tracking of PLHIV health status in light of UNAIDS goals and to estimate the number of HIV-infected persons knowing their condition at a national scale in the upcoming years.

Nonetheless, some limitations of the NDB, with possible consequences on our analysis, are: 1) Due to the Japanese medical insurance reimbursement system, the names of diseases

entered by the physicians are not always definitive, and "possible disease names" tend to be listed. To overcome this limitation, we only considered individuals who received drugs used to treat exclusively HIV-infected patients (not prescribed for diseases other than HIV infection). 2), Duplications could be possible in this anonymized surveillance dataset. 3) The NDB lacked data on patients whose medical expenses are fully paid by the welfare system. However, we believe that the impact of this selection bias is not large, only 6·5% of PLHIV in Japan are eligible for the welfare program [17]. 4) The treatment rate may appear low in prefectures with a small number of PLHIV because the impact of untreated patients becomes more significant.

## Conclusions

This study demonstrates that the Japanese HIV/AIDS medical program, based on the universal health insurance system, and the increasing adoption of highly effective ART with better patient adherence have contributed to positive results. The information stored in large databases, such as the NDB, has the potential to become a valuable tool for tracking the progression of high-burden diseases like AIDS and optimizing health policies.

## Supporting information

**S1 Table. Antiretroviral therapy codes.**
(DOCX)

**S2 Table. The proportion of Fig 3.** (a) Time course of parameters related to PLHIV status in Japan by ratio, 2016–2020. (b) Parameters related to PLHIV status in Japan by ratio, 2020. (c) Parameters related to PLHIV status across the nine Japanese regions by ratio, 2020. (d) Parameters related to PLHIV status according to the mean volume of outpatients per facility in Japan by ratio, 2020. > = 100: Facilities with more than 100 retained patients, 10–99: Facilities with 10–99 retained patients, 1–9: Facilities with 1–9 retained patients.
(DOCX)

**S1 Fig. Trends in antiretroviral therapies throughout the period 2013–2020.** Relevant changes in nucleoside reverse transcriptase inhibitors used (a), key drug classes prescribed (b), and administration regimens selected (c) can be observed during this period. The percentages were calculated by considering the total number of PLHIV receiving antiretroviral treatment. Data source: NDB.
(TIF)

**S2 Fig. Introduction rate of the top ten antiretroviral therapies across Japanese prefectures, 2016–2020.** Data source: NDB.
(TIF)

## Acknowledgments

We are grateful to the Mitsubishi Research Institute, Inc., for their analytical assistance and to the Study Group on the Development of Medical Systems for HIV Infection for providing specialized care to PLHIV in Japan. Editorial support in the form of medical writing, assembling tables, and creating high-resolution images was provided by Editage and Cactus Communications.

## Author Contributions

**Conceptualization:** Yoshiyuki Yokomaku, Tatsuya Noda.

**Funding acquisition:** Yoshiyuki Yokomaku, Tatsuya Noda.

**Investigation:** Yoshiyuki Yokomaku, Tatsuya Noda, Mayumi Imahashi.

**Methodology:** Yuichi Nishioka, Tomoya Myojin.

**Project administration:** Yoshiyuki Yokomaku, Tatsuya Noda.

**Resources:** Yoshiyuki Yokomaku, Tatsuya Noda.

**Supervision:** Aikichi Iwamoto, Tomoaki Imamura.

**Visualization:** Mayumi Imahashi.

**Writing – original draft:** Yoshiyuki Yokomaku, Tatsuya Noda, Mayumi Imahashi.

**Writing – review & editing:** Yoshiyuki Yokomaku, Tatsuya Noda, Mayumi Imahashi, Yuichi Nishioka, Tomoya Myojin, Aikichi Iwamoto, Tomoaki Imamura.

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
