## [Decision Letter · Decision Letter 0]

1 Aug 2024

PONE-D-24-25980Anti-HIV therapies and status of people living with HIV in Japan: An update from hospital survey and National DatabasePLOS ONE

Dear Dr. Noda,

Thank you for submitting your manuscript to PLOS ONE. After careful consideration, we feel that it has merit but does not fully meet PLOS ONE’s publication criteria as it currently stands. Therefore, we invite you to submit a revised version of the manuscript that addresses the points raised during the review process.

We look forward to receiving your revised manuscript.

Kind regards,

Nagarajan Raju

Academic Editor

PLOS ONE

Journal Requirements:

This work was supported by Health Labour Sciences Research Grants 20HB2001 (acquired by YY), 20HB1001 (acquired by TN), and 23HB1001 (acquired by TN) [funder URL: https://mhlw-grants.niph.go.jp/] and by JSPS KAKENHI Grant JP20H00623 (acquired by TN) [funder URL: https://www.jsps.go.jp/english/e-grants/]. The funders played no role in the study design, data collection, data analysis, interpretation, or manuscript writing.

**Additional Editor Comments:**

I suggest authors to go through all the comments from the reviewers and properly address them in the revised version of the manuscript

Reviewers' comments:

Reviewer's Responses to Questions

**Comments to the Author**

1. Is the manuscript technically sound, and do the data support the conclusions?

Reviewer #1: Partly

Reviewer #2: Partly

Reviewer #3: Yes

Reviewer #4: Yes

Reviewer #5: Yes

2. Has the statistical analysis been performed appropriately and rigorously? 

Reviewer #1: Yes

Reviewer #2: No

Reviewer #3: Yes

Reviewer #4: Yes

Reviewer #5: Yes

3. Have the authors made all data underlying the findings in their manuscript fully available?

Reviewer #1: No

Reviewer #2: No

Reviewer #3: Yes

Reviewer #4: Yes

Reviewer #5: Yes

4. Is the manuscript presented in an intelligible fashion and written in standard English?

Reviewer #1: Yes

Reviewer #2: Yes

Reviewer #3: Yes

Reviewer #4: Yes

Reviewer #5: Yes

5. Review Comments to the Author

**Reviewer #1:** This study is about the overall epidemiology and ART use among PLHIV in Japan. It provided a variety of analytical results through a survey of a wide range of hospitals and access to the nationwide database. This is an interesting and necessary study, but I think it needs revision, especially including further clarification on some of the definitions.

Major comments

Abstract

AIDS patients -> PLHIV (also in the manuscript)

Introduction

I think it would be more appropriate to describe the detailed characteristics of NDB in the Methods section rather than in the Introduction section.

Methods

Line 101 Lists of the hospitals that participated in the survey and, if possible, of the respondents are required. It would be good to provide the information as a supplementary table.

Line 127 “Patient retained in care”

Please describe in detail what it means to be linked after diagnosis. For example, is this the same as the definition in the NDB of having visited a hospital or clinic at least once within a 6-month period? Also, I am curious about how cases are handled if PLHIV are transferred to another hospital while receiving treatment at a core hospital.

In addition, please provide a definition for “on treatment.”

Line 145 A more detailed explanation of how PLWHIV are extracted from the NDB is needed. An operational definition for extraction needs to be presented.

Results

Line 174 How is “regularly visit” defined? There is a need to explain.

Line 180 Table 1 You need to check whether “Total number of regular outpatients” and “Number of outpatients” have changed.

Line 221 In Figure 2, there are 5 lines, but only 4 indicators are attached.

Line 235, Line 239 Are CD4 count<200mm3 and AIDS status based on the time of HIV diagnosis?

Line 227-232 I think it would be a good to go to the Methods section.

Line 241-245 How can the number of patients with CD4 count less than 200 be greater than the number of AIDS? An explanation of how AIDS was defined in this study is needed.

Line 308-310 It is necessary to explain in the Discussion section why there are differences between the NDB and core hospital survey results, and why the relative ratios appear high or low depending on the region.

Minor comments

Line 238 "PLHIV patients" should be modified to "PLHIV".

Line 367 The expression “pandemic” does not seem appropriate for HIV infection.

**Reviewer #2:** This study aims to provide updated information on the success of HIV treatment and the care status of AIDS patients in Japan. The analysis utilized structured questionnaire data collected from major hospitals across Japan and data from the National Database of Health Insurance Claims (NDB) from 2016 to 2020. The main findings indicate that over 95% of diagnosed outpatients received continuous treatment, and over 99% were successfully treated.

However, there are several areas in the manuscript that need improvement:

・The current manuscript lacks sophistication in its writing, leading to important points being obscured. Clear and concise presentation of specific data and results is essential for the reader's understanding.

・The objectives and outcome measures of the clinical research are poorly defined. Clear definitions of outcomes such as "treatment success" and "continuous treatment" should be provided, along with detailed results based on these outcomes.

・The abstract should focus on specific data and results. Abstract and general statements, such as "The NDB is a valuable tool for studying the time course of high-burden diseases such as AID S and outlining improved health policies." are unnecessary and should be omitted to keep the abstract concise and impactful.

・The term "core hospitals" appears to be specific to Japan. It is important to clearly explain what constitutes a core hospital, the criteria for their selection, and their role in the healthcare system. This will aid international readers in understanding the context and significance of the data.

**Reviewer #3:** This manuscript is clearly demonstrating the current status of people living with HIV in Japan. Please revise the points below.

1. Page 6, line 122, please add closing parentheses to 4,5, and 6.

2. Page 13, line 290, is preefecture prefecture?

3. Page 14, line 320 to 327, please specify STRs, DVY, TVD, SMT, BVY, and DVT in the main text.

**Reviewer #4:** This manuscript described that updated information on PLHIV in Japan from 2016-2020 using combination data of fore hospital survey and National Data Base(NDB). Authors revealed that more than 95% of diagnosed PLHIV are under treatment and more than 99% were successfully treated with efficient antiretroviral therapy which is mostly single tablet regimen. Large data set were used to conduce this research which would greatly contribute to the future health policy.

There are some points that we think to be more specified. Please see following comments.

・Mehods: Please mention about the excluding criteria for the collected data.

・Table1 showed the annual number of all-cause death and it seems that number is increasing in 2020. Do you have any information about age range, nationality, or disease which caused of death for this data?

・Figure 1,2,３, are not clearly seen. Please make them more visible.

・Figure 2b showed the number of non-Japanese patients were increasing recently. Do you have data of subgroup analysis regarding the rate of sustained treatment and success rate among these population? They may have some difficulties continuing treatment and if there are any difference between Japanese and non-Japanese patients in terms of treatment adherences.

**Reviewer #5:** This study provides valuable data covering a broad range in Japan based on responses from NDB and medical professionals. While the study is generally well-considered, please address the following points:

On line 291, the authors state that "all regions had a treatment introduction rate of ≥94%," which indicates that there are regions where the percentage of patients under treatment falls below 95%. Given that the overall rate exceeds 95%, it can be inferred that regions with rates below 95% likely have fewer patients. Is there a correlation between the small number of patients and the lower proportion of patients under treatment? If possible, please consider adding a discussion that takes into account the regional characteristics.

The vertical axis of Figure 3 shows actual numbers, which makes it difficult to assess the achievement rate of 95-95. If possible, please consider adding a "%" sign to the graph or creating a separate table in the supplementary information.

6. PLOS authors have the option to publish the peer review history of their article (what does this mean?). If published, this will include your full peer review and any attached files.

Reviewer #1: No

Reviewer #2: No

Reviewer #3: No

Reviewer #4: No

Reviewer #5: No

---

## [Author Response · Author response to Decision Letter 0]

24 Oct 2024

Response to Reviewers

First of all, we all thank five reviewers for reviewing our manuscript entitled “Anti-HIV therapies and status of people living with HIV in Japan: An update from hospital survey and National Database.” To facilitate your review of our revisions, the following is a point-by-point response to the questions and comments.

[Reviewer #1]

Major comments

• Abstract

AIDS patients -> PLHIV (also in the manuscript)

Response: We revised all “AIDS patients” to “PLHIV”.

• Introduction

I think it would be more appropriate to describe the detailed characteristics of NDB in the Methods section rather than in the Introduction section.

Response: Thank you for your comment. To focus on the overview and objectives of the NDB, omitting the detailed methods as they were redundant. We moved L66-70 to Method part (Line 140-144) and deleted L71-78.

• Methods

Line 101 Lists of the hospitals that participated in the survey and, if possible, of the respondents are required. It would be good to provide the information as a supplementary table.

Response: Given the significant burden of investigating and providing accurate English translations for all the hospitals, we apologize for the inconvenience, but we will include the URL with the list in Japanese in the Fig 1a legend (Line 206-207). Additionally, the location map is available in Fig. 1a of the paper, which we hope you will find helpful. 

• Line 127 “Patient retained in care”

Please describe in detail what it means to be linked after diagnosis. For example, is this the same as the definition in the NDB of having visited a hospital or clinic at least once within a 6-month period? Also, I am curious about how cases are handled if PLHIV are transferred to another hospital while receiving treatment at a core hospital.

In addition, please provide a definition for “on treatment.”

Response: If diagnosed with HIV, PLHIV will be referred to a medical facility that can provide ART (antiretroviral therapy). The physician at that facility will determine when to start treatment and assist in preparing the necessary documents for applying for a disability certificate, ensuring that financial support is available before the treatment begins. Since the financial support system continues even if the patient transfers to another hospital, PLHIV can continue their treatment with peace of mind.

We provided the definition of “on treatment” in Method part after the definition of “Patient retained in care.” The definition of “Patient on treatment” is HIV-infected person under sustained treatment. 

The NDB tracks all medical visits of the same patient, including hospital transfers. Since patients are tracked using a unique ID, one of the strengths of the NDB is its ability to monitor transfers as well.

Line 145 A more detailed explanation of how PLWHIV are extracted from the NDB is needed. An operational definition for extraction needs to be presented.

Response: We defined patients who had been prescribed anti-HIV drugs (S1 Table) at least once as PLHIV receiving treatment, and identified all the prescription patterns for these PLHIV. Line 147 has been added.

• Results

Line 174 How is “regularly visit” defined? There is a need to explain.

Response: We revised the phrase "regularly visit" to "retained in care".

• Line 180 Table 1 You need to check whether “Total number of regular outpatients” and “Number of outpatients” have changed.

Response: Since the Table 1 might be misunderstanding, we revised the table 1.

• Line 221 In Figure 2, there are 5 lines, but only 4 indicators are attached.

Response: Thank you for your comment. We revised and added the last indicator "AIDS cases (ACTR)".

• Line 235, Line 239 Are CD4 count<200mm3 and AIDS status based on the time of HIV diagnosis?

Response: Yes.

• Line 227-232 I think it would be a good to go to the Methods section.

Response: Thank you for your comment. We moved L227-232 to the Methods section (Line 133-138).

• Line 241-245 How can the number of patients with CD4 count less than 200 be greater than the number of AIDS? An explanation of how AIDS was defined in this study is needed.

Response: Unlike the CDC's definition of AIDS, in Japan, AIDS is defined as the condition where an individual is infected with HIV and has one or more of the 23 AIDS indicator diseases listed below. Thus, PLHIV with a CD4 count of less than 200 but without AIDS-defining illnesses is not diagnosed with AIDS.

"Indicator Diseases”

A. Fungal Infections

Candidiasis (esophageal, tracheal, bronchial, pulmonary)

Cryptococcosis (excluding lung)

Coccidioidomycosis

(1) Disseminated throughout the body

(2) Occurring in locations other than lung, cervical, or hilar lymph nodes

Histoplasmosis

(1) Disseminated throughout the body

(2) Occurring in locations other than lung, cervical, or hilar lymph nodes

Pneumocystis pneumonia

(Note) The classification name for P. carinii has been changed to P. jirovecii

B. Protozoal Infections

6. Toxoplasmic encephalitis (after 1 month of age)

7. Cryptosporidiosis (with diarrhea lasting over 1 month)

8. Isosporiasis (with diarrhea lasting over 1 month)

C. Bacterial Infections

9. Pyogenic bacterial infections (under 13 years of age, with two or more occurrences of the following within 2 years, due to pyogenic bacteria such as Haemophilus or Streptococcus)

(1) Sepsis

(2) Pneumonia

(3) Meningitis

(4) Osteoarticular infection

(5) Abscess in sites other than the middle ear, skin, mucous membranes, or deep-seated organs

10. Salmonella bacteremia (recurrent, excluding typhoid fever)

11. Active tuberculosis (pulmonary or extrapulmonary) (*)

12. Non-tuberculous mycobacterial infection

(1) Disseminated throughout the body

(2) Occurring in locations other than lung, skin, cervical, or hilar lymph nodes

D. Viral Infections

13. Cytomegalovirus infection (after 1 month of age, affecting organs other than liver, spleen, or lymph nodes)

14. Herpes Simplex Virus Infection

(1) Ulcers of mucous membranes or skin lasting over 1 month

(2) With bronchitis, pneumonia, or esophagitis after 1 month of age

15. Progressive multifocal leukoencephalopathy

E. Tumors

16. Kaposi's sarcoma

17. Primary brain lymphoma

18. Non-Hodgkin lymphoma

19. Invasive cervical cancer (*)

F. Others

20. Recurrent pneumonia

21. Lymphocytic interstitial pneumonia / pulmonary lymphoid hyperplasia: LIP/PLH complex (under 13 years of age)

22. HIV encephalopathy (dementia or subacute encephalitis)

23. HIV wasting syndrome (generalized weakness or wasting syndrome)

• Line 308-310 It is necessary to explain in the Discussion section why there are differences between the NDB and core hospital survey results, and why the relative ratios appear high or low depending on the region.

Response: It is believed that the difference arises because, in some regions, the percentage of PLHIV who are visiting core hospitals varies across the area. This contrasts with the results of the NDB, which can track the entire PLHIV population. We added this explanation in the Discussion section (Line 364-367). 

Minor comments

• Line 238 "PLHIV patients" should be modified to "PLHIV".

Response: We revised all “PLHIV patients” to “PLHIV”.

• Line 367 The expression “pandemic” does not seem appropriate for HIV infection.

Response: Thank you for your comment. We revised pandemic to endemic (Line 368).

[Reviewer #2]

• The current manuscript lacks sophistication in its writing, leading to important points being obscured. Clear and concise presentation of specific data and results is essential for the reader's understanding.

Response: Thank you for your advice. We revised our manuscript with proof reading by native English speaker. 

• The objectives and outcome measures of the clinical research are poorly defined. Clear definitions of outcomes such as "treatment success" and "continuous treatment" should be provided, along with detailed results based on these outcomes. 

Response: Instead of using “treatment success” and “continuous treatment”, we changed those to ”2nd and 3rd of the 90-90-90 target” so that readers can easily understand the objectives of this study. 

• The abstract should focus on specific data and results. Abstract and general statements, such as "The NDB is a valuable tool for studying the time course of high-burden diseases such as AIDS and outlining improved health policies." are unnecessary and should be omitted to keep the abstract concise and impactful.

Response: We deleted the indicated part. 

• The term "core hospitals" appears to be specific to Japan. It is important to clearly explain what constitutes a core hospital, the criteria for their selection, and their role in the healthcare system. This will aid international readers in understanding the context and significance of the data.

Response: Core Hospitals were designated by MHLW where ART is provided according to the guidelines. We cited the report previously published about core hospitals by Aikichi Iwamoto et al. (Lines 53-55)

[Reviewer #3]

• Page 6, line 122, please add closing parentheses to 4,5, and 6.

Response: We added closing parentheses to 4, 5, 6.

• 2. Page 13, line 290, is preefecture prefecture?

Response: We corrected the typo.

• 3. Page 14, line 320 to 327, please specify STRs, DVY, TVD, SMT, BVY, and DVT in the main text.

Response: We spelled out all the abbreviations in S1 Table. Please refer to the S1 Table.

[Reviewer #4]

• Methods: Please mention about the excluding criteria for the collected data.

Response: We inserted the following sentence in Method part: Hospitals not informing the number of patients retained after diagnosis, the number of patients on treatment, and the number of patients that were virally suppressed were excluded from our analysis.

• Table1 showed the annual number of all-cause death and it seems that number is increasing in 2020. Do you have any information about age range, nationality, or disease which caused of death for this data?

Response: Unfortunately, we do not have additional data supporting the increase death in 2020.

• Figure 1,2,3, are not clearly seen. Please make them more visible.

Response: As reviewer#4 mentioned, the figures attached to the manuscript were not clear. However, the figures we submitted through PLOS ONE portal site were clear and had high resolution.

• Figure 2b showed the number of non-Japanese patients were increasing recently. Do you have data of subgroup analysis regarding the rate of sustained treatment and success rate among these population? They may have some difficulties continuing treatment and if there are any difference between Japanese and non-Japanese patients in terms of treatment adherences.

Response: Thank you for your valuable feedback. Your question gets to the heart of the matter. Unfortunately, we are unable to obtain information on oral medication adherence from the database. Additionally, we have not conducted any subgroup analyses.

[Reviewer #5]

• On line 291, the authors state that "all regions had a treatment introduction rate of ≥94%," which indicates that there are regions where the percentage of patients under treatment falls below 95%. Given that the overall rate exceeds 95%, it can be inferred that regions with rates below 95% likely have fewer patients. Is there a correlation between the small number of patients and the lower proportion of patients under treatment? If possible, please consider adding a discussion that takes into account the regional characteristics.

Response: As a reviewer indicated, this is the limitation of our study. Thus, we added the limitation in the Discussion part. (Line 398-399)

• The vertical axis of Figure 3 shows actual numbers, which makes it difficult to assess the achievement rate of 95-95. If possible, please consider adding a "%" sign to the graph or creating a separate table in the supplementary information.

Response: To improve visibility, the proportions have been created in the supplemental file. Please refer to S2 Table.

---

## [Decision Letter · Decision Letter 1]

25 Nov 2024

PONE-D-24-25980R1Anti-HIV therapies and status of people living with HIV in Japan: An update from hospital survey and National DatabasePLOS ONE

Dear Dr. Noda,

Thank you for submitting your manuscript to PLOS ONE. After careful consideration, we feel that it has merit but does not fully meet PLOS ONE’s publication criteria as it currently stands. Therefore, we invite you to submit a revised version of the manuscript that addresses the points raised during the review process.

We look forward to receiving your revised manuscript.

Kind regards,

Nagarajan Raju 

Academic Editor

PLOS ONE

Journal Requirements:

Additional Editor Comments:

I suggest authors to address the minor corrections from one of the reviewers

Reviewers' comments:

Reviewer's Responses to Questions

**Comments to the Author**

1. If the authors have adequately addressed your comments raised in a previous round of review and you feel that this manuscript is now acceptable for publication, you may indicate that here to bypass the “Comments to the Author” section, enter your conflict of interest statement in the “Confidential to Editor” section, and submit your "Accept" recommendation.

Reviewer #1: All comments have been addressed

Reviewer #2: All comments have been addressed

Reviewer #3: All comments have been addressed

Reviewer #4: All comments have been addressed

2. Is the manuscript technically sound, and do the data support the conclusions?

Reviewer #1: Yes

Reviewer #2: Partly

Reviewer #3: Yes

Reviewer #4: Yes

3. Has the statistical analysis been performed appropriately and rigorously? 

Reviewer #1: Yes

Reviewer #2: No

Reviewer #3: Yes

Reviewer #4: Yes

4. Have the authors made all data underlying the findings in their manuscript fully available?

Reviewer #1: Yes

Reviewer #2: Yes

Reviewer #3: Yes

Reviewer #4: Yes

5. Is the manuscript presented in an intelligible fashion and written in standard English?

Reviewer #1: Yes

Reviewer #2: Yes

Reviewer #3: Yes

Reviewer #4: Yes

6. Review Comments to the Author

Reviewer #1: I think the authors have responded faithfully to the comments, and there are no additional comments from me.

Reviewer #2: Thank you for your revisions; however, after reviewing the updated manuscript, several key areas remain insufficiently addressed, affecting clarity and accessibility for an international audience.

Despite proofreading, the manuscript still lacks the level of clarity and sophistication required for effective communication. Critical points remain unclear, which limits readers’ ability to understand the main conclusions of the study. Additional refinement is necessary to ensure a clear, precise, and engaging presentation of the findings.

While you replaced "treatment success" and "continuous treatment" with "90-90-90 target" terminology, this does not clarify the outcome measures. Using this terminology without explicitly defining terms, such as “treatment success” and “continuous treatment,” makes it difficult for readers to interpret the outcomes. Providing precise definitions within the study context would help avoid confusion.

Although the general phrase "the NDB is a valuable tool" was removed, the abstract still lacks conciseness and specificity. It would benefit from focusing exclusively on specific data and results, without general statements, to enhance clarity and impact.

The term "core hospitals" remains unclear for an international readership. Relying solely on a Japanese-language source to explain the role and criteria of these hospitals limits understanding. It is essential to provide a thorough explanation of the significance and selection criteria of these institutions within the healthcare system. This context would make the manuscript more accessible and meaningful for non-Japanese readers.

Reviewer #3: The manuscript has been well revised according to the points below.

•Page 6, line 122, please add closing parentheses to 4,5, and 6.

Response: We added closing parentheses to 4, 5, 6.

•2. Page 13, line 290, is preefecture prefecture?

Response: We corrected the typo.

•3. Page 14, line 320 to 327, please specify STRs, DVY, TVD, SMT, BVY, and DVT in

the main text.

Response: We spelled out all the abbreviations in S1 Table. Please refer to the S1

Table.

Reviewer #4: Dear Dr. Noda

I have thoroughly reviewed the manuscript titled " Anti-HIV therapies and status of people living with HIV in Japan: An update from hospital survey and National Database" submitted to PLOS ONE. After careful consideration, I believe the paper provides valuable insights and contributes to the current understanding of HIV therapies in Japan.

7. PLOS authors have the option to publish the peer review history of their article (what does this mean?). If published, this will include your full peer review and any attached files.

Reviewer #1: No

Reviewer #2: No

Reviewer #3: No

Reviewer #4: No

---

## [Author Response · Author response to Decision Letter 1]

24 Dec 2024

Please refer to the Response to Reviewers2 file.

---

## [Decision Letter · Decision Letter 2]

3 Jan 2025

Antiretroviral therapies and status of people living with HIV in Japan: An update from hospital survey and National Database

PONE-D-24-25980R2

Dear Dr. Noda,

We’re pleased to inform you that your manuscript has been judged scientifically suitable for publication and will be formally accepted for publication once it meets all outstanding technical requirements.

Kind regards,

Nagarajan Raju

Academic Editor

PLOS ONE

Additional Editor Comments (optional):

Reviewers' comments:

Reviewer's Responses to Questions

**Comments to the Author**

1. If the authors have adequately addressed your comments raised in a previous round of review and you feel that this manuscript is now acceptable for publication, you may indicate that here to bypass the “Comments to the Author” section, enter your conflict of interest statement in the “Confidential to Editor” section, and submit your "Accept" recommendation.

Reviewer #1: All comments have been addressed

Reviewer #3: All comments have been addressed

2. Is the manuscript technically sound, and do the data support the conclusions?

Reviewer #1: Yes

Reviewer #3: Yes

3. Has the statistical analysis been performed appropriately and rigorously? 

Reviewer #1: Yes

Reviewer #3: Yes

4. Have the authors made all data underlying the findings in their manuscript fully available?

Reviewer #1: Yes

Reviewer #3: Yes

5. Is the manuscript presented in an intelligible fashion and written in standard English?

Reviewer #1: Yes

Reviewer #3: Yes

6. Review Comments to the Author

Reviewer #1: (No Response)

Reviewer #3: There is no additional point to be revised.

My comments were bwlow.

•Page 6, line 122, please add closing parentheses to 4,5, and 6.

Response: We would like to thank Reviewer 3 for reviewing our manuscript again and for

providing comments, which have considerably helped us improve our manuscript.

Based on your comment, we have added closing parentheses to 4, 5, 6.

•2. Page 13, line 290, is preefecture prefecture?

Response: The typographical error has been corrected now.

•3. Page 14, line 320 to 327, please specify STRs, DVY, TVD, SMT, BVY, and DVT in

the main text.

Response: As per your comment, we have spelled out all the abbreviations in S1 Table.

7. PLOS authors have the option to publish the peer review history of their article (what does this mean?). If published, this will include your full peer review and any attached files.

Reviewer #1: No

Reviewer #3: No

---

## [Editor Report · Acceptance letter]

16 Jan 2025

PONE-D-24-25980R2 

PLOS ONE

Dear Dr. Noda, 

I'm pleased to inform you that your manuscript has been deemed suitable for publication in PLOS ONE. Congratulations! Your manuscript is now being handed over to our production team.

Kind regards, 

on behalf of

Dr. Nagarajan Raju 

Academic Editor

PLOS ONE